# Lineal Discrimination of Horses and Mules. A Sympatric Case from Arauca, Colombia

**DOI:** 10.3390/ani10040679

**Published:** 2020-04-13

**Authors:** Arcesio Salamanca-Carreño, Jordi Jordana, Rene Alejandro Crosby-Granados, Jannet Bentez-Molano, Pere M. Parés-Casanova

**Affiliations:** 1Grupo de Investigaciones los Araucos, Facultad de Medicina Veterinaria y Zootecnia, Universidad Cooperativa de Colombia, Arauca 810001, Colombia; asaca_65@yahoo.es (A.S.-C.); rene.crosby@campusucc.edu.co (R.A.C.-G.); jannet.bentez@campusucc.edu.co (J.B.-M.); 2Department de Ciència Animal i dels Aliments, Universitat Autònoma de Barcelona, 08193 Bellaterra, Spain; jordi.jordana@uab.es; 3Departament de Ciència Animal, Universitat de Lleida, 25198 Lleida, Spain

**Keywords:** biometrics, hybrids, morphologic traits, body measurement

## Abstract

**Simple Summary:**

The recognition of morphological differences between horses and mules has received not much attention, although this topic has a great interest, between others, in zooarchaeological research. Moreover, from an ethnological point of view, local type mules have been rarely described. This investigation aimed to compare morphologically mules and horses and also to describe mule from Arauca region, in Colombia. Morphological quantitative traits for Araucan mules are presented here for first time, and moreover it is demonstrated that some postcranial anatomical elements can give enough information for a bone differentiation between horses and mules from this region. Proposed discrimant formula possibly must be changed in other areas where both species were sympatric.

**Abstract:**

This is the first morphological comparative study between local horses and mules from Arauca, Colombia. It was realized to compare morphological traits between both species by analysing 15 adult mules (7 males and 8 females) and 150 adult horses (137 males and 13 females), with an age interval from 2 to 22 years. Data consisted of 24 different body quantitative traits which can explain the body conformation: thoracic circumference, body length (BL), thoracic depth and width, withers height (WH), sternum height, shoulders width, chest width, forelimb cannon perimeter and length, head length and width, skull length and width, face length and width, ear length and width, loin height, croup height (CrH), width and length, dock height (DoH), and hock height. Heart girth circumference, body length, withers height, croup height, and dock height were the most discriminative traits, showing statistical differences between species. The formula is X = (BL × 0.402) + (WH × 0.323) + (CrH × 0.352) + (DoH × 0.384). A value of X > 184.5 assigns with total certainty that a skeleton belongs to a horse, and if X < 174.0, it is a mule. The proposed formula has a 100% specificity but a 71.4% sensibility for mules and an 84.4% for horses into the rank of 174.0–184.5. Therefore, results demonstrate that some postcranial anatomical elements of *Equus* could give enough information for a bone differentiation between horses and mules, at least in animals from the Araucan region, but the main interest is that it reflects the possibility to differentiate morphometrically both species from bone remains when horses and mules were sympatric.

## 1. Introduction

Body conformation is an important parameter in phenotypic classification of any breed or population [1] and it can reflect stringent environmental of extreme areas, as exist in Arauca, East Colombia. In this sense, the body shape of an animal population can be viewed as determined by its biological functionality and productive use [1,2,3].

Mules and horses have traditionally been part of extensive pastoral systems in Arauca, providing an essential transport, pack, and draught resource as working animals [4]. Horses are believed to have descended from horses first brought from Spain during the 15th century and were selected for adaptation to the local conditions [4]. These extremely hardy and well-adapted horses (*Equus caballus*) and their hybrids (male donkey × female horse) were very important during the colonization for transportation of goods and people for raising livestock and, if necessary, for meat [4]. In relation to the equine hybrids, it is worth mentioning that the different species of Equidae have different diploid numbers of chromosomes [5]. Domestic horses have a chromosome number of 64 and donkeys have that of 62, leading to mules having 63 chromosomes [4]; therefore their hybrid offspring have an odd number of chromosomes, resulting in the vast majority of these animals being sterile [4].

Many morphological reports are available on Araucan horse [6,7,8], and practically none are available on mules. In another aspect, mules are often not mentioned in a positive way in the zooarchaeological literature, as they cannot often easily be identified. The problem lies in the fact that mules are thought to be only trivially osteologically different from horses, but perhaps, there are more numbers than supposed of unaccounted mules in archaeological records [4]. 

This is the first study in literature focusing on mule morphological traits and comparing body measurements of native mules raised in Arauca, East Colombia. As management for horse and mules in Arauca is totally equal and they occupy the same territory, they are sympatric. Biases due to ecological, nutritional, or handling factors can be excluded. However, this research not only provides descriptions on Araucan domestic local equids but also intends to find morphological differences between horses and their hybrids, focusing mainly on those traits clearly supported by a bony basis and inferring osteological remnants. If the outlined separation would be achieved, then it will be possible to address an accurate picture of remains in osteological remains in areas where both species coexist or have coexisted, as not necessarily our results can be inferred to other areas.

## 2. Materials and Methods

### 2.1. Experimental Animals

In this study, a total of adult 15 mules (hybrids between a mare and a jackass, 7 males and 8 females) and 150 adult horses (137 uncastrated males and 13 females) were considered. Animals belonged to 16 different farms (“fincas”) of the Department of Arauca, East Colombia: Acacias, Arenosa, Buenos Aires, Bahía, Belencito, Cabañas, Chenchena, Delicias, El Secreto, La Paz, Libertad, Mercedes, Nueva Vida, Ranchito, Victoria, and Zamuracos, and all were considered to be of Araucan type [8].

Animals were measured with a specially graduated measuring tape and a rigid ruler. Ages were determined from the information given by owner. Data consisted of 24 different body quantitative traits which can explain the body conformation: thoracic circumference (ThP), body length (BL), thoracic depth (ThD) and width (ThW), withers height (WH), sternum height (StH), shoulders width (ShW), chest width (ChW), forelimb cannon perimeter (CaP) and length (CaL), head length (HdL) and width (HdW), skull length (SkL) and width (SkW), face length (FaL) and width (FaW), ear length (EaL) and width (EaW), loin height (LoH), croup height (CrH), width (CrW) and length (CrL), dock height (DoH), and hock height (HoH) (Appendix A). These traits considered are those traditionally encompassed for standard morphological breed studies, and detailed anatomical reference points can be found there [9]. To overcome the effect of age, only animals above 2 years were included in the study, and final age interval ranged from 2 to 22 years.

### 2.2. Statistical Analysis

First, a two-way NPMANOVA (Nonparametric Multivariate Analysis of Variance) with species (horse and mule) and sex as factors was done, with 9999 permutations and Euclidean distances. Principal Component Analysis (PCA) was then used to evaluate the traits, considering species and sex separately. It was obtained from a var-covar matrix. Alternatively, a one-way NPMANOVA was performed with the considered most informative traits from PCA as well as a multivariate regression analysis to age. Finally, a discriminant analysis was used to generate a function to differentiate between species.

Analysis were done with PAST v. 2.17c software (Oslo, Norway) [10]. *p* values less than 0.05 were considered as statistically significant.

## 3. Results and Discussion

The two-way ANOVA reflected differences for both species (horse and mule) and sex, but their interaction was not significative (Table 1). Genders were not globally different if we consider species separately (*p* > 0.1). PCA showed that PC1 explained 93.9% of the total observed variance and that PC2 explained 3.8% (Table 2), although the placement of individuals in the ordination space showed no clear separation between them (Figure 1). Variation in the data is mainly related to size (heights) differences. Heart girth circumference (ThP), body length (BL), withers height (WH), croup height (CrH), and dock height (DoH) were the most discriminative traits. Surprisingly, ear length (EaL) was not a very discriminative trait.

One-way NPMANOVA for those 5 most discriminative traits showed statistical differences between species (*F*_4, 820_ = 522.9, *p* < 0.0001). In Table 3, there appear descriptive statistics for those values, which were statistically higher for horses. None of them showed a significative correlation to age (Wilk’s λ = 0.955, *F*_5, 159_ = 1.488, *p* = 0.196) (graph not showed here). Probably from a biometrical point of view, mules are smaller in Arauca due to the small size of jackasses and a low uterine volume of mares.

If we consider 4 most discriminative traits which show statistical differences between species and have no dependence on age on the studied range, excluding thoracic circumference which is impossible to calculate on skeletons (due to the muscular cingula which surrounds the thoracic perimeter), the following formula can be stablished:X = (BL × 0.402) + (FaW × 0.323) + (CrH × 0.352) + (DoH × 0.384)(1)

A value of X > 184.5 assigns with total certainty that a skeleton belongs to a horse, and if X < 174.0, it is a mule. Intermediate rank (174–184.5) is not informative. The proposed formula has a 100% specificity but a 71.4% sensibility for mules and 84.4% for horses (e.g., 6 mules and 27 horses were erroneously classified in that intermediate rank).

It is unreasonable to use this as a guide for differentiating horses and mules with complete skeletons for animals from other breeds or geographical origins and if a researcher does not dispose of entire and articulated skeletons. However, at least, our results demonstrate that postcranial skeleton of pure and hybrid *Equus* can give enough information for differentiating rather accurately between horses and mules in sympatric areas.

## 4. Conclusions

Some postcranial measurements can give enough information for skeleton differentiation between sympatric horses and mules from Araucan plains. Although proposed discrimant formula possibly must be changed in other areas, obtained results demonstrate that postcranial skeleton measurements of pure and hybrid *Equus* can give enough information for differentiating rather accurately between horses and mules.

## Figures and Tables

**Figure 1 animals-10-00679-f001:**
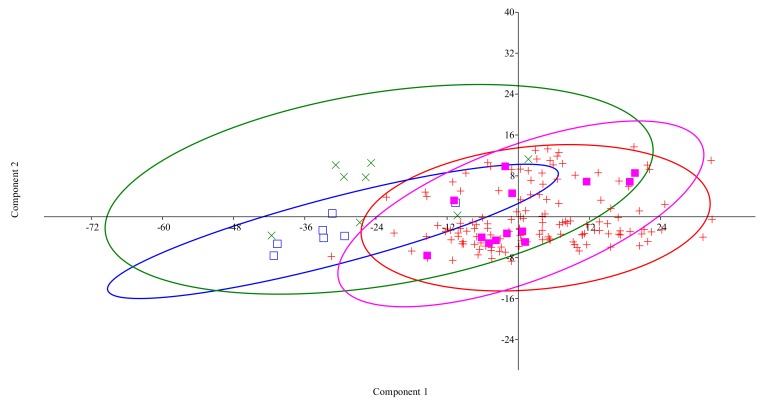
Principal component analysis plot run on the measured set of biometrical variables, with samples (individuals) separated by sex and species for adult 15 mules (7 males □ and 8 females *x*) and 150 adult horses (137 uncastrated males + and 13 females ■): First Principal Component (PC1) explained 93.9% of the total observed variance, and PC2 explained 3.8%. The placement of individuals in the ordination space showed no clear separation between them.

**Table 1 animals-10-00679-t001:** Two-way NPMANOVA (Nonparametric Multivariate Analysis of Variance) for both species and sex for adult 15 mules (7 males and 8 females) and 150 adult horses (137 uncastrated males and 13 females): Analysis reflected significative differences between sexes and species but not their interaction. F-test is the mean square for each main effect and the interaction effect divided by the within variance. The interaction effect is the effect that one factor has on the other factor.

Source	Sum of Squares	Degrees of Freedom	Mean Square	F	*p*
Species	12,052	1	12,052	10.337	0.0001
Sex	2276.6	1	2276.6	1.952	0.0017
Interaction	−1.26 × 10^5^	1	−1.26 × 10^5^	−107.69	0.9641
Residual	1.88 × 10^5^	161	1166		
Total	76,492	164			

**Table 2 animals-10-00679-t002:** Loading values for Principal Component 1 (PC1) explained 93.9% of the total observed variance. PC2 explained 3.8%. Heart girth circumference (ThP), body length (BL), withers height (WH), croup height (CrH), and dock height (DoH) were the most discriminative traits for PC1 (values > 0.3, which appear in bold).

Traits Quantitative	Axis 1	Axis 2
EaL	−0.14250	0.2225
EaW	−0.05775	0.0526
SkW	−0.04198	0.2319
SkL	−0.00793	0.1814
HdW	−0.00117	0.0998
FaW	0.00885	0.0413
HdL	0.03741	0.0614
CaL	0.03953	0.2605
FaL	0.04534	−0.1199
CaP	0.05945	0.2807
ChW	0.07713	0.5706
ThD	0.08367	0.1072
ShW	0.09077	0.1934
HoH	0.14720	−0.1662
CrL	0.16360	0.2012
WH	0.16670	0.1041
ThW	0.20710	0.1822
LoH	0.22420	0.1246
CrW	0.22730	−0.0086
**WH**	**0.32380**	0.0342
**CrH**	**0.35280**	0.1966
**DoH**	**0.38400**	0.0094
**BL**	**0.40290**	−0.2650
**ThP**	**0.42960**	−0.2689

ThP: thoracic circumference; BL: body length; ThD: thoracic depth; ThW: thoracic width; WH: withers height; StH: sternum height; ShW: shoulders width; ChW: chest width; CaP: forelimb cannon perimeter; CaL: forelimb cannon length; HdL: head length; HdW: head width; SkL: skull length; SkW: skull width; FaL: face length; FaW: face width; EaL: ear length; EaW: ear width; LoH: loin height; CrH: croup height; CrW: croup width; CrL: croup length; DoH: dock height; HoH: hock height.

**Table 3 animals-10-00679-t003:** Main descriptive statistics for traits which were significatively different between horses (*n* = 150) and mules (*n* = 15) (*p* < 0.001). See text for acronyms. Lineal values in cm. Coefficient of variation in %.

Horses	ThP	BL	WH	CrH	DoH
Min	137	117.4	119.3	123	105.5
Max	177	147	154.5	146.5	134
Mean	154.2	131.5	133.9	135.1	121.7
Stand. deviation	6.9	6.0	5.2	4.8	5.3
Coefficient of variation	4.5	4.6	3.9	3.6	4.3
Mules					
Min	132	114.5	115.8	117	105
Max	164	128	133	136	117.5
Mean	140.3	119.6	123.8	124.8	110.5
Stand. deviation	8.7	3.7	5.4	5.4	4.0
Coefficient of variation	6.2	3.1	4.4	4.4	3.6

ThP: thoracic circumference; BL: body length; WH: withers height; CrH: croup height; DoH: dock height.

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
