# Peer review of "Lineal Discrimination of Horses and Mules. A Sympatric Case from Arauca, Colombia"

_animals, 2020, doi:10.3390/ani10040679_

Round 1

Reviewer 1 Report

The article provides a statistical analysis of the results of linear measurements of horses (n=150) and mules (n=15) from the Arauka of Colombia to show anatomical differences between these animal species. Of the many parameters ultimately, four of them were taken into account to output the differentiating factor: ALC (withers height), LCO (body lenght), ALGR (croup height) and ANC (dock height). No exact methodology (anatomical points) for these values was given.The X = (LCO*0.402)+(ALC*0.323)+(ALGR*0.352)+(ANC*0.384) is only the sum of linear parameters expressed in cm and is proportional to the size of the animal. It can be assumed that the smaller the animal is the lower the factor and vice versa. It follows that the smallest animals of the test population n=165 are mule (X<174cm), the biggest are horses (X>184,5cm) and among the averages it is impossible to establish unequivocally the species affiliation of an individual on the basis of the calculation of the X factor.It would be more reasonable to look for a different relationship between the individual's parameters than the sum. Althought such differences can now be easily demonstrated in geneticstudies (e.g.different chromosomes) it is worth looking for traditional solutions. The method presended will not find the wider use, as there are many breeds of horses that will have an index X smaller than 174cm, which is the indicator that the authors say is dedicated to the mules population.

Author Response

It has been stated that results are only effective among animals from Arauca region, although is reflects interesting insights into bone domestic Equus differentiation. Abreviations in the tables have been clarified.

Reviewer 2 Report

General Comments: The English terms to Spanish abbreviations are hard to follow ~ it requires writing an abbreviation list just to remember what the abbreviations referenced.  These need to be translated.

Materials and Methods:

  • What were the breed of horses used across this study? How does this compare to the pedigree of the mules?  How translational would this study be when comparing the skeletons of different breeds of horses?
  • Line 86: p-value correction for multiple testing?
  • If the formula can be used to differentiate 100% of mules and only just over half of the horses in the study, then how many of these horses and mules fell into the indiscriminate category? E. what is the sensitivity and specificity of using this formula in the field?  Machine learning to discriminate between species?
  • The authors need to include a discussion/conjecture as to why these four measurements are different between species.
  • How do Colombia mules translate to other mules across regions (i.e. how translational is this study outside of Columbia)?
  • There is no reference to the supplemental material in the manuscript, and I cannot evaluate the supplemental material as it is not translated.

Author Response

(The authors gave the same response as above.)

Round 2

Reviewer 1 Report

The explanations provided by the authors regarding the impact of the economic factor on the implementation of the undertaken research are convincing. It is always better to do something in a given case than to do nothing because of a lack of funds. The assumed goal was achieved by much longer path, but at the same time the authors characterized the population of horses and mules in detail.

Author Response

Thanks.

Reviewer 2 Report

At this point, the authors have not addressed all of the concerns with this manuscript:

Question: Line 86: p-value correction for multiple testing?

Author Response: yes, why not?

Reviewer Response to Authors: Correct, I am asking for you to clarify this as you used several models.  One of which you performed simulations and the others of which you do not mention whether you performed simulations and whether you opted to do additional correction.

Question: If the formula can be used to differentiate 100% of mules and only just over half of the horses in the study, then how many of these horses and mules fell into the indiscriminate category? E. what is the sensitivity and specificity of using this formula in the field? Machine learning to discriminate between species?

Author Response: the main goal, as it is stated, is that both species can be morphologically differentiated, at least in sympatric areas.

Reviewer Response to Authors: This response does not address the issue here.  Yes, you state that the species can be morphologically differentiated but you need to report the number of individuals that were in either category, particularly the indiscriminate zone.  I.E. if 5 mules fell into this category and 50% of the horses then what value does this truly have in the field?  Over 50% of the time you would not be able to differentiate between the species and this needs to be addressed.

Author Response

Response: it has been stated into the manuscript (Abstract and Results sections).